# Genetic impacts on within-pair DNA methylation variance in monozygotic twins capture gene-environment interactions and cell-type effects

Xiaopu Zhang[1]*, Idil Yet[1,2], Sergio Villicaña[1], Juan Castillo-Fernandez[1], Massimo Mangino[1], Jouke Jan Hottenga[3,4,5], Pei-Chien Tsai[1], Josine L. Min[6], Mario Falchi[1], Andrew Wong[7], Dorret I. Boomsma[3,4,5], Ken K. Ong[8,9], Jenny van Dongen[3,4,5] and Jordana T. Bell[1]*

*Correspondence:
xiaopu.1.zhang@kcl.ac.uk;
jordana.bell@kcl.ac.uk

[1] Department of Twin Research and Genetic Epidemiology, King's College London, London, UK
Full list of author information is available at the end of the article

## Abstract

**Background:** Genetic variants that are associated with phenotypic variability, or variance quantitative trait loci (vQTLs), have been detected for multiple human traits. Gene-environment interactions can lead to differential phenotypic variability across genotype groups, therefore, genetic variants that interact with environmental exposures can manifest as vQTLs. Although changes in DNA methylation variability have been observed in several diseases, vQTLs for methylation levels (vmeQTL) have not yet been explored in depth.

**Results:** We optimize the value of monozygotic twin studies to identify and replicate vmeQTLs for blood DNA methylation variance at 358 CpGs in 988 adult monozygotic twin pairs from two European twin registries. Over a third of vmeQTLs capture identical vmeQTL-environmental factor interactions in both datasets, and the majority of interactions are observed with blood cell counts. Correspondingly, over 60% of CpGs affected by genotype-monocyte and genotype-T cell interactions replicate as CpGs affected by genetic effects in the relevant cell type in an independent dataset. Most vmeQTLs also replicate in 1,348 UK non-twin adults and show longitudinal stability in a sample subset. Integrating gene expression and phenotype association results identifies multiple vmeQTLs that capture GxE effects relevant to human health. Examples include vmeQTLs interacting with blood cell type to influence DNA methylation in *FAM65A*, *NAPRT,* and *CSGALNACT1* underlying immune disease susceptibility and progression.

**Conclusions:** Our findings identify novel genetic effects on human DNA methylation variability within a unique monozygotic twin study design. The results show the potential of vmeQTLs to identify gene-environment interactions and provide novel insights into complex traits.

**Keywords:** DNA methylation, DNA methylation variability, Variance methylation quantitative trait loci, VmeQTL, Gene-environment interactions

## Background

Genome-wide association studies (GWASs) have identified thousands of genetic effects on human traits, where genotypes affect the mean levels of the trait assuming that each genotype group has the same phenotypic variance [1, 2]. However, differences in trait variance across different genotype classes have been observed in multiple phenotypes, including in human disease [3, 4], animal and plant developmental traits [5–9], and phenotypic plasticity in changing environments [10–12]. Specifically, the phenotypic variance of certain allele carriers differs from that of alternative allele carriers at a single locus. Genetic variants which describe such situations are associated with phenotypic variability and manifest as variance quantitative trait loci (vQTLs). The differential phenotypic variance across genotype classes could arise due to differential sensitivity to environmental exposures. Therefore, one important driver of vQTLs is the interactions of the vQTL genetic variants with environmental factors (GxE), or with other genetic variants (GxG) [13–16]. The presence of such GxG and GxE interactions can lead to differences in phenotypic variability across genotypes at a single locus [13, 17–20]. Therefore, identifying vQTLs could uncover genetic variants involved in GxG or GxE interactions, without *a priori* knowledge of the interacting factors. Thus, vQTLs can provide novel insights into the genetic architecture of human complex traits.

Multiple vQTLs have been detected across a range of human phenotypes [13, 15, 17, 21, 22], where follow-up analyses have identified potential GxG and GxE effects underlying the vQTLs. For example, Brown et al. [13] identified 508 vQTLs for gene expression levels, which led to further exploration of 256 signals that captured putative GxG interactions. More recently, Wang et al. [23] detected 75 vQTLs for nine phenotypes in the UK Biobank dataset, of which 7 signals captured potential vQTLs interactions with age that influence bone mineral density, BMI, waist circumference, and basal metabolic rate. These examples illustrate the benefits of vQTLs to uncover statistical GxG or GxE interactions underlying human traits. At present, a systematic genome-wide GxG and GxE detection remains challenging. Most studies identifying direct interaction effects require a clear prior hypothesis about the potential interacting environmental or genetic factor, and some also restrict analyses to a subset of candidate environmental exposures (GxE) or genomic regions (GxG) [3, 24–26]. In contrast, vQTL identification has provided a good alternative for genome-wide detection of potential GxG and GxE interactions in large-scale genomic studies.

DNA methylation is an important mechanism in development, ageing, and disease [19, 20, 24, 27]. Inter-individual differences in DNA methylation are influenced by both genetic and environmental factors [24, 25, 28]. Multiple previous studies have detected differential DNA methylation variability in human disease, for example, compared to the disease-free individuals, greater DNA methylation variability has been observed in patients with type 1 diabetes [26], rheumatoid arthritis [29], cervical intraepithelial neoplasia [30], precursor cancer lesion [31], and multiple cancers [32]. One potential explanation for these observations could be the presence of GxE interactions that would manifest as differential DNA methylation variability across genotype groups [33–35]. While great progress has been made in detecting genetic effects on average DNA

methylation levels, or DNA methylation quantitative trait loci (meQTLs) [28, 36, 37], there are limited studies of genetic effects on DNA methylation variability, or variance DNA methylation quantitative trait loci (vmeQTLs). To our knowledge, Ek et al. [38] carried out the most comprehensive vmeQTL study in humans to date, focusing on local genetic effects in blood samples from 729 unrelated individuals. Ek et al. identified several hundred CpGs affected by vmeQTLs, and a handful of CpG sites were involved in GxE interactions [38]. However, the sample size was relatively modest, suggesting limited power to detect vmeQTLs, and the method used to detect vmeQTLs showed high false positive rate in subsequent independent studies [23, 39, 40].

We sought to identify vmeQTLs to explore evidence for GxE effects on the human blood methylome, using a study design that maximises the value of monozygotic (MZ) twins [41, 42]. MZ twins are ideal to study environmental factor effects, because MZ twins are genetically nearly identical, matched for age and sex, and have more similar early life environments, including prenatal environment, compared to singletons or siblings. In our approach we considered the difference in DNA methylation levels between two genetically identical twins in a MZ twin pair as a measure of DNA methylation variability, where each MZ twin pair was a unit of analysis. vmeQTLs were then identified as genetic variants associated with the intra-twin DNA methylation discordance. The rationale for using this approach is that it eliminates one source of false positive vQTLs in singletons, which is attributed to linkage disequilibrium (LD) [43]. SNPs in partial LD with a mean-level controlling QTL are likely to manifest as spurious vQTLs. Although setting a stringent LD cut-off could reduce the detection of such spurious signals, it will not eliminate them. However, the MZ twin design circumvents this problem, because the two twins in a MZ twin pair have identical germline genetic variation.

In the current study, we applied a within MZ twin pair study design to 988 MZ twin pairs to identify and replicate novel genetic effects on DNA methylation variance. Many of the identified vmeQTLs showed evidence for GxE interactions effects, especially with blood cell subtypes, and a proportion replicated as CpGs affected by meQTLs in the corresponding blood cell type in an independent dataset [44]. Integrating these data with gene expression and phenotype association results, we highlight multiple examples of putative blood cell-related genetic effects on DNA methylation that were highly relevant to human disease.

## Results

We applied vmeQTL detection to whole blood Infinium HumanMethylation450 BeadChip profiles measured from 355 adult female pairs of MZ twins (age range 19–80) from the TwinsUK cohort and from 633 adult MZ twin pairs (69.8% female, age range 18–79) from the Netherlands Twin Register (NTR) (Fig. 1). Using a MZ twin study design we identified and replicated 331 whole blood vmeQTLs that affect DNA methylation variances at 358 vCpGs (CpG sites associated with vmeQTLs). We explored the genomic distribution of vmeQTLs and vCpGs, their putative functional impacts, tissue specificity, longitudinal stability, and replication in non-twin individuals. vmeQTLs-vCpG associations were also tested for genotype-environment interactions in both TwinsUK and NTR datasets with factors including BMI, smoking, and blood cell counts. We sought to replicate putative vmeQTLs interactions with blood cell subtypes using published

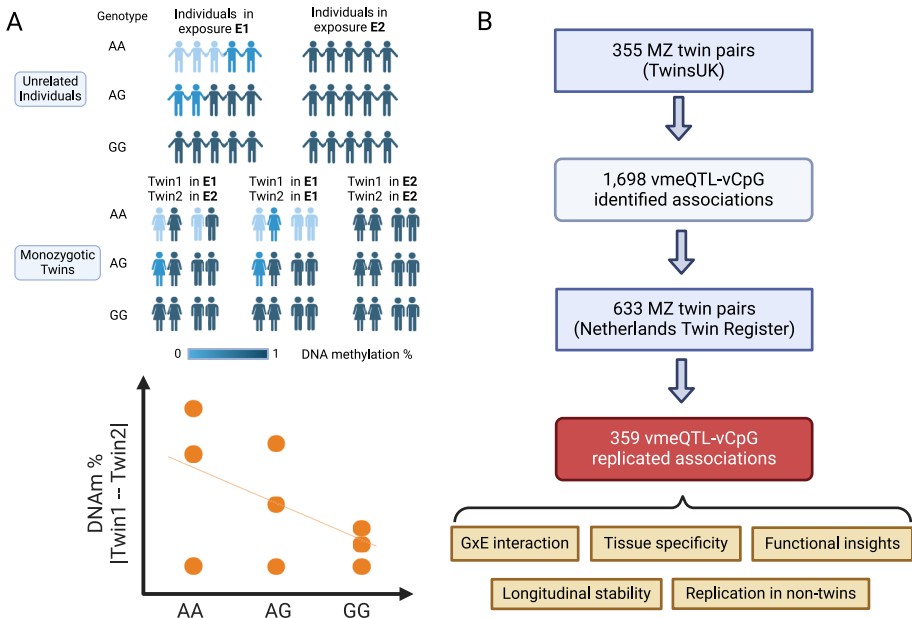

**Fig. 1** Study design. **A** Study hypothesis. The hypothesis is that genetic effects on intra-MZ-twin pair methylation difference manifest as vmeQTLs. Furthermore, vmeQTLs may capture genotype by exposure interaction effects on DNA methylation levels. **B** Study workflow. Genome-wide vmeQTL identification was carried out in TwinsUK and replication was conducted in the Netherlands Twin Register. Follow-up analysis aimed to explore the features of vmeQTLs and vCpGs, as well as their relevance to human complex traits

blood cell-type meQTL results from the Blueprint project [44]. Integrating our results with gene expression, and genome- and epigenome-wide association study (GWAS and EWAS) findings, we identified specific examples of vmeQTLs involved in GxE interactions in the context of several human diseases.

### vmeQTL detection methods in monozygotic twins

To gauge the most optimal method for vmeQTL detection in MZ twin pairs, we used simulated data to compare the performance of six vmeQTL mapping models (Twins1-6). The six models fit different measures of the intra-twin pair phenotype difference as a function of the genotype, with or without considering within-twin pair average phenotype levels. The six models (Twins1-6) included (Twins1): $res(abs) \sim genotype$; (Twins2): $res(abs)^2 \sim genotype$; (Twins3): $res(abs)/res(mean) \sim genotype$; (Twins4): $res^2(abs)/res^2(mean) \sim genotype$; (Twins5): $res(abs) \sim genotype + res(mean)$; (Twins6): $res^2(abs) \sim genotype + res(mean)$.

Here, res(abs) and mean(abs) represent the residuals of the intra-twin pair phenotype difference and of the within-twin pair average phenotype level after regressing out covariates, respectively (see Methods). We assumed that the phenotype was DNA methylation beta values of a single CpG-site, and that this phenotype was affected by a single genetic variant, a single environmental factor, their interaction, and random noise simulated from different distributions. When DNA methylation levels follow the normal distribution, the false positive rates (FPRs) of all six models are near to or lower than 0.05. However, when the phenotype is non-normally distributed, models Twins5 and Twins6 showed high FPR (Additional file 1: Fig. S1A), and in the presence of GxE interactions

($\alpha_{GxE}$.0) models Twins3 and Twins4 exhibit low discovery rates (Additional file 1: Fig. S1B, C). Therefore, models Twins1 and Twins2 perform best and their discovery rates increase with increasing $\alpha_{GxE}$ (see Methods, Additional file 1: Fig. S1A-C). Furthermore, Twins1 performs better than Twins2 when the vQTL effect size ($\alpha_{GxE}$) is below 15%, making the Twins1 model optimal in these simulation settings. Similar results were obtained when using simulated DNA methylation M values (Additional file 1: Methods, Fig. S2).

Lastly, a hurdle in detecting vQTL effects is that the average methylation level and variance are strongly correlated, therefore, if a genetic variant is a mean-controlling QTL it could also be detected as a spurious vQTL. In our simulations we observed that the correlation between the intra-twin difference and the within-twin average level of DNA methylation at the same CpG could reach 0.5 (Spearman correlation, Additional file 1: Fig. S3A) when only additive genetic effects exist without interactions. Under this scenario, the discovery rate of the Twins1 method is less than 5%, suggesting that the probability that the Twins1 method will detect the meQTL as a spurious vmeQTL signal is less than 5%.

### Genome-wide vmeQTL detection in MZ twins from TwinsUK

We initially carried out genome-wide vmeQTL and meQTL analyses in the discovery set of 355 MZ twin pairs from the TwinsUK cohort. For local genetic effects we carried out conditionally independent analyses using tensorQTL [45]. At an FDR 5%, there were 2,338 CpGs (vCpGs) affected by *cis* vmeQTLs, and 51,846 CpGs (mCpGs) affected by *cis* meQTLs. Altogether, 94% (2,208) of TwinsUK vCpGs were also mCpGs, that is, these CpGs were associated with both *cis* vmeQTLs and meQTLs (Additional file 1: Fig. S4). Distal genetic effects were identified using MatrixEQTL [46] with genome-wide significance thresholds estimated by permutations. At an FDR 5%, there were 659 vCpGs affected by *trans* vmeQTLs (nominal $P < 1.08e-11$) and 43,282 mCpGs affected by *trans* meQTLs (nominal $P < 4.5e-8$), where again the majority of vCpGs (62%, 407 CpGs) were also affected by meQTLs (Additional file 1: Fig. S4).

We carried out several tests and sensitivity analyses to verify the robustness of these initial results (Additional file 1: Method, Fig. S4). First, we excluded one vCpG signal that overlapped previously published imprinted regions [47], because our focus is on vmeQTLs that are likely to reflect GxE interactions. Second, we removed putative false positive vmeQTLs that may arise from situations where SNPs in partial LD with mean-controlling QTLs may be associated with methylation variance. For each of the CpGs that are affected by both vmeQTLs and meQTLs simultaneously (2,230 CpGs for *cis* and 460 CpGs for *trans* effects), we conducted LD clumping of all SNPs that were identified as vmeQTLs and meQTLs (Additional file 1: Methods). For a given CpG, if the vmeQTL and meQTL SNPs both fell in the same LD clump, and the meQTL signal was strongest, the vmeQTL SNP was discarded. As a result, we removed vmeQTLs for 1,316 vCpGs in *cis* and 3 vCpGs in *trans* (Additional file 1: Fig. S4). Furthermore, we tested whether the remaining vCpGs with concurrent vmeQTLs and meQTL effects still showed significant vmeQTL-vCpG associations when the meQTL effect was included as a covariate (for 1,021 CpGs for *cis* and 656 CpGs for *trans* in total), due to correlation between DNA methylation mean level and variance (Additional file 1: Method, Fig. S3). As a result, a

further 5 vmeQTL-vCpG associations were filtered out. Finally, we carried out a sensitivity analysis to estimate the effect of including additional covariates for vmeQTL detection (see Methods), and all vmeQTL-vCpG associations remained significant at FDR 5%. In total, the TwinsUK discovery samples identified 1,698 vmeQTL-vCpG associations including 1,665 unique vCpGs after all quality control checks (Additional file 1: Fig. S4).

### vmeQTL-vCpG replication in the Netherlands Twin Register

We sought to replicate the 1,698 vmeQTL-vCpG associations in 633 pairs of MZ twins (442 female pairs and 191 male pairs) from the Netherlands Twin Register (NTR). After quality control data filtering (see Methods), 1,317 TwinsUK vmeQTL-vCpG associations were available for testing in the NTR cohort. We found that 38.4% (339 of 917) of *cis* vmeQTL-vCpG and 20 *trans* vmeQTL-vCpG associations replicated at Bonferroni correction for multiple testing (Fig. 2A). Altogether, 359 vmeQTL-vCpG associations were replicated in the combined female and male NTR twin replication sample (Bonferroni corrected $P < 0.05$).

Given that all twins in the TwinsUK discovery samples are female, we also carried out a sensitivity analysis where replication was pursued only in 442 pairs of female MZ twins from the NTR. We observed that 276 associations surpassed Bonferroni correction for replication in the female-specific analysis, and all 276 associations were also contained in the 359 sex-shared replicated signals (Additional file 2: Table S1), suggesting that the effects are independent of sex. We therefore focused on the 359 replicated vmeQTL-vCpG associations for downstream analysis.

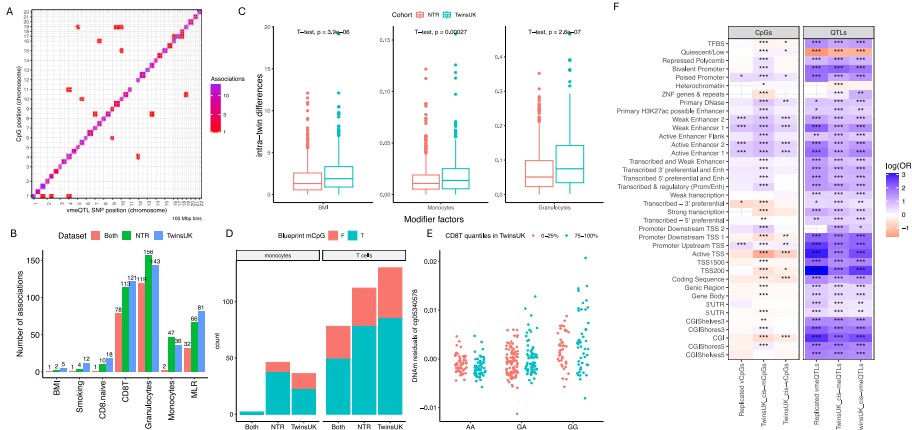

**Fig. 2** Genomic distribution and GxE interaction analysis of vmeQTL-vCpG associations. **A** Genomic distributions of 359 replicated vmeQTL-vCpG associations. **B** Summary of the number of vmeQTL-vCpG associations involved in GxE interactions with different modifier factors in TwinsUK and NTR. **C** Comparison of the intra-twin-pair difference in modifier factors in TwinsUK and NTR. **D** Comparison of vCpGs with GxE effects in monocytes and T-cells, with CpGs affected by meQTLs (mCpG) in the Blueprint project from Chen et al. [44]. The y-axis shows the number of detected CpGs (vCpGs) underlying vmeQTL-cell type interactions in the twin cohorts. The colour of the bar shows which of these CpGs were detected as mCpGs in the corresponding cell type in Blueprint data. **E** GxE example of G - CD8+T cell interaction involving rs6667048 and cg05340578 in the TwinsUK dataset; this GxE interaction example is also captured in NTR. **F** Genomic distributions of detected CpGs and QTLs in the twin data, using genomic annotation categories from Encode, Epigenome Roadmap, and RefSeq

### GxE interactions underlying vmeQTL-vCpG associations identify predominantly cell-type meQTL effects

We explored evidence for potential interactions between the replicated vmeQTLs and 7 modifier variables including BMI, smoking, 4 blood cell components (naïve CD8 + T cells, CD8 + T cells, granulocytes, and monocytes), and blood myeloid to lymphoid ratio (MLR). GxE interaction analyses were carried out using linear mixed-effect additive genetic models, corrected for family relatedness, where the unit of analysis was the twin individual. Of the 359 replicated associations, 208 (57.9%) show evidence for at least one interaction effect after multiple testing correction (FDR < 0.05) in the TwinsUK dataset. These included 5 vmeQTL-BMI, 12 vmeQTL-smoking, and 399 vmeQTL-cell type interactions (Additional file 2: Table S2). In the NTR dataset, 171 out of 359 associations (47.6%) showed evidence for at least one GxE interaction (FDR < 0.05), including 2 vmeQTL-BMI, 4 vmeQTL-smoking, and 292 vmeQTL-cell interactions (Additional file 2: Table S2). Altogether, there were 234 identical GxE interactions identified in both TwinsUK and NTR datasets, that underlie 131 vmeQTLs-CpG associations (37% of replicated vmeQTL effects) (Fig. 2B). Of these, one association result is an interaction with smoking (rs966107 and cg23625390) and one with BMI (rs7487166 and cg13235087), and the remaining 232 results are interactions with blood cell components or myeloid to lymphoid ratio (MLR) (Fig. 2B, Additional file 2: Table S2).

We explored the consistency in distribution of the modifier factors (E in GxE) across the two twin cohorts. To this end, we compared the intra-twin-pair differences of the modifier factors between TwinsUK and NTR. We observed significant differences in the intra-twin-pair discordance for granulocytes (T-test, $P = 2.6e-7$), BMI (T-test, $P = 3.9e-6$), and monocytes (T-test, $P = 2.7e-4$) (Fig. 2C), but not for CD8 T + cells, naïve CD8 T + cells, and rate of smoking concordance or discordance (Additional file 1: Fig. S5).

vmeQTLs involved in genotype by blood cell count interactions for DNA methylation levels may in fact be blood cell type meQTLs. Therefore, we compared our results with cell mCpGs, which were previously identified in monocytes and T cells in Blueprint project data by Chen et al. [41]. From the 359 replicated associations, there were 36 vCpGs with underlying vmeQTL-monocyte interactions in the TwinsUK dataset. Of these, 61% (22 of 36) were also mCpGs specifically in monocytes in Blueprint [44]. Similarly, 65.4% (85 of 130) of vCpGs with underlying vmeQTL-T cell (CD8 + T cells and naïve CD8 + T cells) interactions in the TwinsUK dataset were also mCpGs in T cells in Blueprint data [44]. In the NTR dataset, the corresponding rates of vCpGs influenced by vmeQTL-monocyte and vmeQTL-T cell interactions being cell-type mCpGs in Blueprint monocytes and T cells, are 80.4% (37 of 46) and 69.6% (78 of 112), respectively (Fig. 2D). If we only consider identical vmeQTL-cell type interactions detected in both TwinsUK and NTR, then 62.8% (49 of 78) of vCpGs affected by vmeQTL-T cell interactions were mCpGs in T cells in the Blueprint data (Fig. 2D-E). Only 2 vmeQTL-monocyte interactions were identically detected in the two twin cohorts, potentially due to the significant difference in intra-twin-pair discordance in monocyte levels across the two twin cohorts. However, both of these vCpGs with vmeQTL-monocyte interactions were previously detected as mCpGs in monocytes in Blueprint data (Fig. 2D). We conclude that overall, over 60% of GxE effects involving monocytes, naïve CD8 + T cells, and CD8 + T cells replicate as cell-type genetic effects on DNA methylation in Blueprint data.

### Characteristics and genomic distribution of vCpGs and vmeQTLs

We explored the characteristics and genomic annotations of the 359 replicated vmeQTL-vCpGs associations. There was only 1 vCpG associated with 2 independent *cis*-vmeQTL, and the remaining vCpGs were influenced by a single lead *cis*- or *trans*-vmeQTL. Therefore, the 359 associations include 358 unique vCpGs and 331 unique vmeQTLs. We observed that 95% of *cis*-associated and 100% *trans*-associated vCpGs were also affected by meQTLs. Similarly, 91% and 100% of *cis* and *trans*-vmeQTLs manifest as meQTL for the same CpG, respectively.

We explored the distribution of DNA methylation levels at vCpGs. We found that 320 (89.1%) vCpGs had either hyper- (beta > 0.8) or hypo-methylated (beta < 0.2) DNA methylation levels. We also observed that the intra-twin-pair DNA methylation difference across the 358 vCpGs is larger than that of the remaining 261,404 tested CpGs (mean(intra-twin pair methylation difference of vCpGs) = 0.02, range = 0.002–0.08; mean(intra-twin pair methylation difference of non-vCpGs) = 0.019, range = 0.0004–0.19; one-sided t-test $P$ = 0.01). However, the mean level of DNA methylation at the 358 vCpGs is not significantly different from that of non-vCpGs (mean(mean methylation level of vCpGs) = 0.51; mean(mean methylation level of non-vCpGs) = 0.54; two-sided t-test $P$ = 0.12).

We next considered the genomic distribution of the vmeQTLs and vCpGs (Fig. 2F). Consistent with some patterns previously reported for meQTLs [36, 48, 49], the replicated vmeQTLs SNPs are depleted in quiescent regions (odds ratio (OR) = 0.20, 95% CI [0.16, 0.24], $P$ = 2.07e-42)), but are enriched in many regulatory genomic regions, especially in CpG islands (OR = 14.8, 95% CI [9.84, 21.6], $P$ = 2.91e-24) and transcription start sites (TSS) (harmonic mean OR = 13.09). In terms of CpGs, the replicated vCpGs with vmeQTLs also show a similar distribution to that of mCpGs [36, 48, 49] (Fig. 2F), and are enriched in poised promoters (OR = 1.66, 95% CI [1.1, 2.42], $P$ = 1.19e-2), enhancers (harmonic mean OR = 2.00), and in the promoter region upstream of the TSS (OR = 1.61, CI [1.28, 2.02], $P$ = 5.27e-5), but are depleted in transcribed regions (OR = 0.56, CI [0.33, 0.9], $P$ = 1.39e-2). However, unlike mCpGs, we did not observe a significant enrichment or depletion of vCpGs with respect to CpG density, gene regions, and other functional genomic regions (Fig. 2F).

With regard to transcription factor binding sites (TFBSs), the replicated vmeQTLs show enrichment in TFBSs when grouping all TFBSs in a single category. If we consider individual TFBSs, vmeQTLs were enriched in some TFBSs (for example, CTCF) while vCpGs were depleted in other TFBSs (for example, KLF5) (Additional file 1: Method, Fig. S6). However, vCpGs do not show significant enrichment or depletion for TFBSs overall. We also compared vCpGs to three sets of published metastable epialleles (MEs) from previous studies [50–52], and none of the vCpGs were previously reported as MEs.

### vmeQTL effects show longitudinal stability

To explore the robustness of vmeQTL effects, we next examined whether vmeQTL effects are longitudinally stable in a subset of 56 MZ twin pairs from the 355 discovery TwinsUK MZ twin pairs. Blood samples were collected at two time points 4 to 16 years apart (Additional file 2: Table S3), where the first time-point was included in the

discovery set in the current study (Illumina 450K) and was also profiled for DNA methylation using the Infinium MethylationEPIC array (Illumina EPIC). The second-time point blood DNA methylation profile was generated by the Illumina EPIC array only. We therefore used only Illumina EPIC data at both time points in these longitudinal analyses. Of the 359 replicated vmeQTL-vCpG associations from the Illumina 450K results, 136 vCpGs were removed because they showed weak correlation between Illumina 450K and EPIC profiles at time point 1 (Spearman's rank correlation < 0.2, Additional file 2: Table S4). Therefore, the longitudinal analyses focused on 223 vmeQTL-vCpG associations in paired twin samples.

We observed that the intra-twin DNA methylation differences across MZ twin pairs were not associated with time gap intervals for all 223 tested vCpGs (FDR < 0.05, Fig. 3A). Furthermore, we tested whether the vmeQTL effects were also stable over time. Of the 223 tested vmeQTL-vCpG associations, 172 associations were significant at timepoint 1 (FDR < 0.05, Fig. 3B). This reduction is likely attributed to differences in array

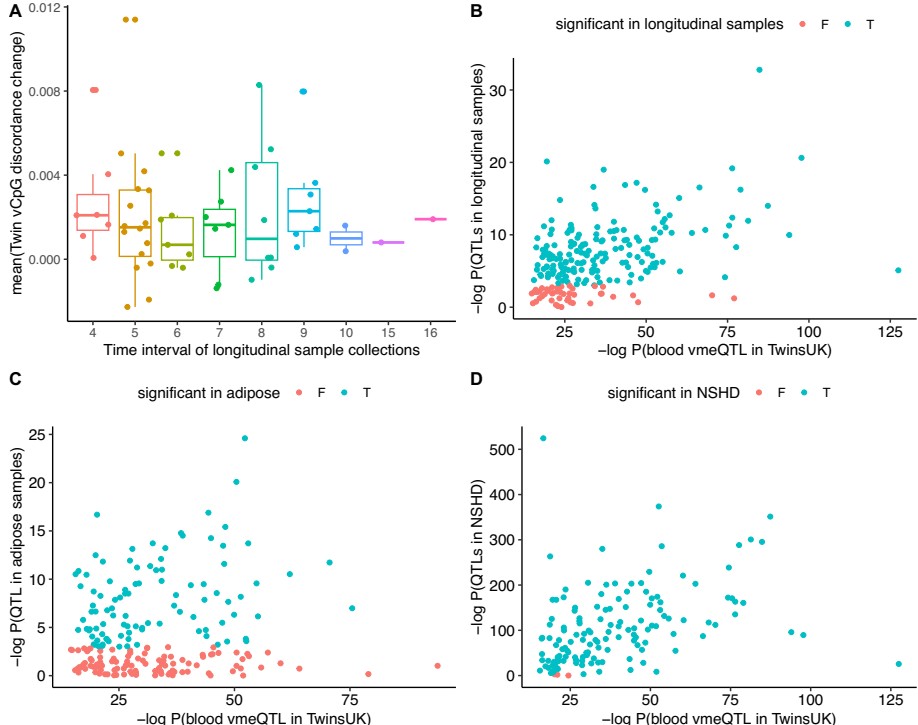

**Fig. 3** Characterizing vmeQTL-vCpG effects longitudinally, across tissues, and in non-twin individuals. **A** Longitudinal intra-MZ-twin pair methylation difference over time. The Y axis represents the average discordance of intra-MZ-twin pair methylation difference of vCpGs, and the X-axis shows the time interval (years) of the longitudinal sample collection for each twin pair. The methylation data of the longitudinal samples were measured by the EPIC array. The X-axis in panels B-D shows the *P*-values of the 450K blood vmeQTL in the TwinsUK discovery set. **B** vmeQTL longitudinal stability. The Y-axis shows *P*-values of vmeQTLs at the second-time point blood samples for the longitudinal samples. Altogether, 153 of 172 tested associations remained significant at timepoint 2 (in blue). **C** A proportion of blood vmeQTLs validate in adipose tissue. The Y-axis shows the P-value of the 450K adipose vmeQTLs in the TwinsUK dataset of 85 MZ twin pairs. Altogether, 82 of 207 tested associations are significant in adipose tissue (in blue). **D** Replication of vmeQTLs in non-twins. The Y-axis shows the P-values of the DRM method to detect blood vmeQTLs in 1,348 non-twin individuals from the NSHD cohort. Altogether, 193 of 200 tested associations are significant (in blue). The NSHD methylation data were measured by the EPIC array

profiling technology (450K vs EPIC), and also to the much smaller sample size (56 versus 355 MZ twin pairs in the discovery set), resulting in differences in power and allele frequency. Of the 172 time point 1 significant associations, 153 (89%) remain significant at time point 2 at an FDR of 5%. These results suggest that the majority of vmeQTL effects are longitudinally stable.

### Tissue specificity of vmeQTL-vCpG associations

meQTLs exhibit both tissue-specific and tissue-shared effects [28]. Therefore, we investigated whether the vmeQTL-vCpG associations detected in blood show evidence of tissue specificity, by estimating their effect on adipose tissue DNA methylation profiles. Adipose tissue samples were obtained from 85 pairs of female MZ twins from the TwinsUK cohort [53, 54], where 56 MZ twin pairs overlapped with the blood vmeQTL discovery dataset. After data filtering, 207 out of 359 vmeQTL-vCpG associations were available to test in the adipose tissue dataset (see Methods). We observed that 39.6% (82 vmeQTL-vCpGs) of the associations demonstrated significant vmeQTL-vCpG effects in adipose tissue at an FDR 5% (Fig. 3C). Given that adipose tissue also contains immune cells, the validated associations may represent vCpGs affected by vmeQTL-immune cell interactions. Indeed, 34 of the 82 adipose vmeQTL-vCpG associations captured 73 GxE interactions in TwinsUK blood samples, including 71 vmeQTLs-cell type interactions and 2 vmeQTL-BMI interactions. Consequently, these results suggest that a substantial fraction of the genetic effects on DNA methylation variability in blood could be shared across tissues.

### Replication of vmeQTL-vCpG associations in non-twin individuals

The 359 vmeQTL-vCpG associations were identified and replicated in MZ twin pairs, but given previous reports of MZ-twin specific DNA methylation variation [55], we sought to ascertain whether these associations also persist in non-twin individuals. We explored replication of the 359 vmeQTL-vCpG associations in 1,348 non-twin individuals from the MRC National Survey of Health and Development (NSHD) or 1946 UK birth cohort [56, 57], using Illumina EPIC blood DNA methylation profiles. Based on the Illumina 450K and EPIC array correlation results in the subset of 56 MZ twin pairs in TwinsUK, we only considered 200 (out of 359) vmeQTL-vCpG associations to test in NSHD. To test for vmeQTL effects on vCpGs in unrelated individuals, we applied the deviation regression model (DRM) [39] and we also used the squared residual value linear model (SVLM) [58] as a sensitivity analysis. Altogether, 193 of 200 vmeQTL-vCpG associations were significant at FDR 5% in the NSHD data when applying the DRM approach (Fig. 3D). SVLM validated 180 (90%) associations at FDR 5%, all of which were detected by DRM (Additional file 2: Table S5). These replication findings in non-twin individuals suggest that vmeQTLs are robust in twin and non-twin samples and are independent of MZ twin-specific features.

### Functional and phenotypic insights from vmeQTLs and vCpGs

We explored the genes that the replicated vCpGs mapped to, to gain potential functional insights. First, we performed pathway analysis of the 324 genes that were annotated to vCpGs using FUMA [59], compared to genes annotated to ~261,000

tested CpGs as background [59]. We observed enrichments in cellular molecular functions, such as peptide and antigen binding, and pathways involved in immune diseases, for example, in autoimmune thyroid disease, asthma and type 1 diabetes (Additional file 2: Table S6). Second, we explored potential functional impacts of the vCpGs by comparing them to CpGs correlated with gene expression in external datasets. To this end, we compared the 358 vCpGs to expression quantitative trait methylation loci (eQTM) previously identified in the BIOS consortium dataset [60]. We observed that vCpGs are significantly enriched to also be eQTMs (OR = 3.97, 96%, CI [2.79, 5.51], $P = 2.23e$-12), and specifically, 41 vCpGs were identified as *cis* eQTMs to nearby genes (Additional file 2: Table S7).

We next explored if the vmeQTLs and vCpGs forming the 359 vmeQTL-vCpG replicated associations have previously been reported to have genetic or epigenetic associations with human phenotypes and diseases in published GWAS and EWAS. Strikingly, 353 of 358 unique vCpGs were previously identified as EWAS signals [61]. The predominant EWAS traits included age (292 vCpGs), tissue (257 vCpGs), ancestry (66 vCpGs), and gestational age (47 vCpGs) [61]. We carried out enrichment analyses to assess whether these EWAS results were more or less likely expected by chance. We found significant enrichments for vCpGs to be EWAS signals for cardiometabolic, immune, and neuropsychiatric traits, as well as for diseases such as bladder cancer and cystic fibrosis, and for exposures such as malathion and household socioeconomic status in childhood (Fig. 4).

We also explored if vmeQTL SNPs are more likely to fall in GWAS loci. When considering all genetic variants that surpass the vmeQTL detection threshold, we observed an enrichment of vmeQTLs in GWAS signals for cancers, such as squamous cell lung carcinoma, immune diseases, such as sarcoidosis, and multiple traits including neuroticism and aspects of brain morphology (Fig. 4). Notably, we found that vmeQTL enrichment for certain GWAS trait loci may depend on individual contexts. For example, vmeQTLs are depleted in GWAS signals for estimated glomerular filtration rate (eGFR) (OR = 0.36, 95% CI [0.20, 0.61], $P = 9.3e$-6), but are enriched in GWAS signals for eGFR in individuals with diabetes (OR = 7.46, CI [2.47, 19.4], $P = 3.8e$-4) (Fig. 4). Therefore, this suggests that although vmeQTLs are not associated with eGFR in the general population, vmeQTLs could impact eGFR in individuals with diabetes, and thus may be specific to diabetes-associated eGFR signals. Similarly, we observed the enrichment of vmeQTLs (OR = 3.27, 95% CI [2.8, 3.8], $P = 7.4e$-37) in lung cancer GWAS signals, but a depletion of vmeQTLs (OR = 0.70, 95% CI [0.54, 0.89], $P = 2.7e$-3) in lung cancer GWAS signals detected in ever smokers only. This implies that these vmeQTLs may display distinct effects on DNA methylation according to smoking status, as well as different impacts on lung cancer risk. This aligns with the interpretation that vmeQTLs show different effects on DNA methylation according to specific environmental exposures. These findings provide insights into different regulatory pathways underlying human complex traits, particularly when genetic effects on a phenotype may depend on environmental exposures.

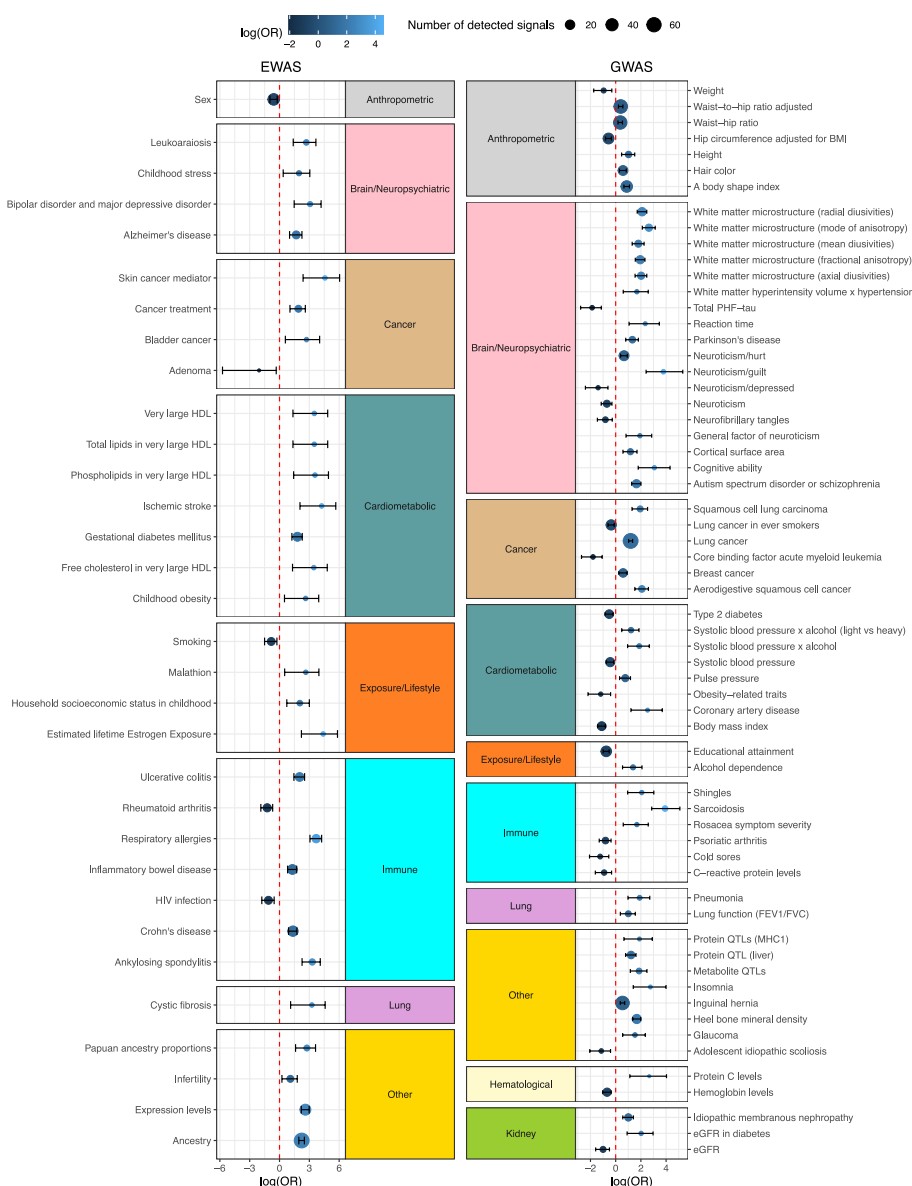

**Fig. 4** Enrichment of vCpGs and vmeQTLs in EWAS and GWAS loci. EWAS and GWAS results are grouped according to phenotype area in 10 broad phenotypic categories. The Y-axis shows the EWAS or GWAS traits for which vCpGs or vmeQTLs surpass enrichment to fall in the corresponding EWAS or GWAS loci. The X-axis shows enrichment or depletion results. Results are shown at FDR 5%, after adjusting *P*-values for two-sided enrichment analysis

## Examples of vmeQTL of relevance to human traits and disease

The enrichment of vmeQTLs and vCpGs in GWAS and EWAS signals led us to explore specific examples of vmeQTL-vCpG associations in complex traits. We selected vmeQTL-vCpG associations that capture putative GxE interactions, where both the vmeQTLs were GWAS signals and the vCpGs were EWAS signals, and also the vCpGs were previously associated with gene expression levels as eQTMs in the BIOS dataset [60].

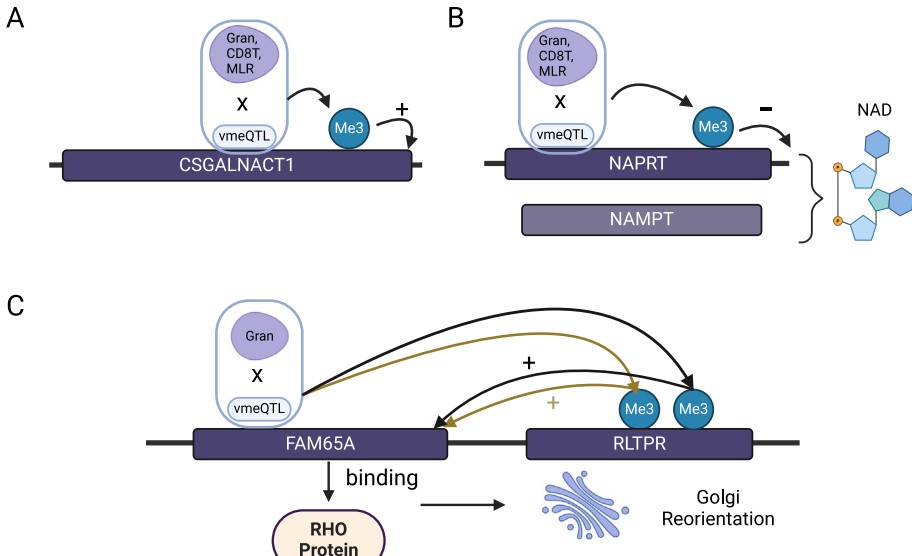

**Fig. 5** Examples of vmeQTL-vCpG associations relevant to human disease. Each example shows the location of the vmeQTL and vCpG in the gene, the interacting modifier factor (in purple), and the direction of methylation correlation with gene expression (positive or negative). **A** Interaction between vmeQTL rs7833274 with granulocytes (Gran), CD8 + T cells (CD8T), and MLR influences methylation levels at cg00923880 in *CSGALNACT1*; previously linked to immune phenotypes and disease, and skeletal and joint development. **B** Interaction between vmeQTL rs896955 and granulocytes, CD8 + T cells, and MLR influences DNA methylation levels at cg16316162 in *NAPRT*; *NAPRT* and Nicotinamide phosphoribosyltransferase (*NAMPT*) are two major intracellular enzymes in the biosynthesis of NAD, which is relevant to immune disease and inflammatory response. **C** Interaction between vmeQTL rs9934328 and granulocytes influences DNA methylation levels at cg01291088 and cg09316954 which influence the expression of *FAM65A*; previously linked to immune diseases and cancers

One example is the *cis* vmeQTL-vCpG association between rs7833274 (chr8:19,613,134) and cg00923880 (chr8:19,613,178) (Fig. 5A). DNA methylation variability of cg00923880 is associated with interactions between rs7833274 and granulocytes, CD8 T cells, and MLR. Both rs7833274 and cg00923880 are located in the *CSGALNACT1* gene, which is vital in skeletal and joint development. Variant rs7833274 is associated with appendicular lean mass [62]. Lower methylation level of cg00923880 has been linked to higher risk of respiratory allergies [63] and systemic lupus erythematosus (SLE) [64, 65], as well as lower expression of *CSGALNACT1* [60]. Previous findings show that children with allergy often present with musculoskeletal disease [66]. SLE patients often suffer from joint pain [67], and are at increased risk of developing osteoporosis and avascular necrosis of bone and osteomyelitis compared to the general population [68]. Furthermore, immune cells and proinflammatory cytokines affect the functions of osteoclasts and osteoblasts that lead to bone loss [68]. This example reveals intricate regulatory networks linking immune and bone disease, underscoring the complexity of these regulatory interactions. Therefore, the captured putative cell-specific effects of rs7833274 on DNA methylation levels at cg00923880 in *CSGALNACT1* may provide novel insights into mechanisms underlying development and progression of respiratory allergies and SLE, and biological pathways linking immune and musculoskeletal disease.

Another example is *cis*-vmeQTL rs896955 (chr8:144,658,370) and vCpG cg16316162 (chr8:144,660,157), which uncover interactions between rs896955 and granulocytes,

CD8 T cells, and MLR that are associated with DNA methylation levels of cg16316162 in both TwinsUK and NTR datasets (Fig. 5B). Hypermethylation of cg16316162 has been detected in primary Sjogren's syndrome [69], which is an autoimmune disease characterized by inflammation. Both rs896955 and cg16316162 are located in the gene encoding nicotinate phosphoribosyltransferase (*NAPRT*). In the BIOS dataset, greater methylated levels at cg16316162 are correlated with reduced expression of *NAPRT*, which is one of two major intercellular enzymes in the biosynthesis of nicotinamide adenine dinucleotide (NAD) [70]. NAD is important in regulating cellular functions, including inflammatory response [69, 70]. Therefore, one hypothesis derived from our results is that the effects of rs896955 on the methylation level of cg16316162 depend on blood cell composition, which in turn may impede the production of NAD in specific blood cell subtypes. This disruption in NAD production may affect the inflammatory response, and as a result, increase risk of primary Sjogren's syndrome.

Lastly, rs9934328 (chr16:67573367) is a GWAS signal of haemoglobin levels and is associated in *cis* with vCpGs cg01291088 (chr16:67687648) and cg09316954 (chr16:67687754) (Fig. 5C). rs9934328 is located in gene *FAM65A* while both cg01291088 and cg09316954 are located in gene *RLTPR* and associated with the expression level of *FAM65A*. *FAM65A* encodes Rho family-interacting cell polarization regulator 1, which binds to RHO proteins to reorient the Golgi apparatus during cell migration, leading to alterations in cell polarity and function [71]. *FAM65A* is overexpressed in various diseases, including cancers, where Golgi reorientation plays an important role in aberrant cell proliferation and survival during cancer metastasis [68–70]. Previous findings indicate that hypermethylation of cg01291088 and cg09316954 is observed in patients with respiratory allergy and inflammatory bowel disease (IBD), respectively [63, 72]. Anaemia is common in patients with IBD, with an estimated one third of IBD cases presenting with low haemoglobin levels [73]. In our data rs9934328-granulocyte interactions are associated with DNA methylation levels at both cg01291088 and cg09316954, where higher methylation levels are linked to increased *FAM65A* expression, as well as a higher risk of IBD. These interaction effects suggest that understanding the role of *FAM65A* in immune diseases and their complications may require consideration of the cellular environment, specifically granulocyte blood cell subtypes.

## Discussion

The current study detects genetic effects on DNA methylation variability with a MZ twin design. In total, we identified and replicated 359 pairs of vmeQTL-vCpG associations, of which over a third capture identical GxE interaction effects on DNA methylation in the two independent twin cohorts. Our subsequent analyses show that the majority of vmeQTL effects are longitudinally stable and can be validated in non-twin samples. Comparison with GWAS and EWAS signals indicates that vmeQTL and vCpG signals show enrichment for associations with human diseases. Integration of vmeQTLs with results from GxE interactions, methylation-gene expression comparisons, and GWAS and EWAS signals, allows us to present several examples of disease-relevant vmeQTLs where our results highlight specific aspects of cellular or organismal environment that may contribute to disease development, disease progression, or development of complications.

Over a third (37%) of vmeQTLs signals captured identical GxE interactions in both twin cohorts, where in the majority of cases the environmental exposure was cellular environment, specifically, blood cell subtypes. Although we restricted analyses to GxE interaction effects that were identified by both TwinsUK and NTR datasets, we actually observed more GxE interactions in each cohort independently with over 57.9% and 47.6% of vmeQTLs capturing GxE effects in TwinsUK and NTR, respectively. A potential reason for the lower rate of replicated GxE interactions across the two twin cohorts (37%), is that some of the modifier factors exhibit different distributions across the two cohorts. We explored these results at the level of intra-twin pair differences of the modifier factors, and identified differences for monocytes, granulocytes, and BMI. Furthermore, G-cell subtype interactions account for the largest variability of DNA methylation in each cohort. Over 60% of G-cell subtype interaction effects involving monocytes, naïve CD8 + T cells, CD8 + T cells validated as cell type mCpGs in Blueprint data, which suggests that these interaction effects with cell profiles are biologically meaningful. Comparison of the vmeQTLs with EWAS signals showed that the predominant overlapping signals were obtained for EWAS of age, tissue, ancestry, and gestational age, all of which are also related to changes in blood cell types [74–77]. Distinct blood cell type proportions can also be a marker of different diseases or specific environmental exposures that may not be directly measured in our sample. Therefore, our result of G-cell interactions underlying vmeQTLs may also reflect interactions with other unmeasured environmental factors that are correlated with changes in blood cell subtypes. In addition, our results indicate that specific vmeQTLs often capture G-cell interactions with several different blood cell subtypes. This suggests that vmeQTLs, rather than being cell specific mQTLs, can affect multiple (but not all) blood cell subtypes. By looking at the pattern of cell subtype effects we may also be able to infer the developmental timing of the specific methylation mark during haematopoietic cell lineage establishment.

Gene-environment interactions have been widely reported to affect human traits and diseases. For example, interactions between cigarette smoking and genetic variation can lead to a decrease in lung function [78]. Here, we further integrated EWAS, GWAS, eQTM signals with the captured GxE interaction results to explore potential links between vmeQTL-vCpG associations and specific human complex traits. We provide several examples relevant to immune health such as primary Sjogren's syndrome, IBD, respiratory allergies and systemic lupus erythematosus. Our results suggest that the allele dosage of vmeQTLs can have a varied effect on DNA methylation levels depending on blood cell type, and therefore have different influences on disease-related vCpGs. This is furthermore important to explore in human diseases because changes in blood cell counts can manifest as predictors of risk and mortality for several diseases, for example, white blood cells are independent predictors of cardiovascular disease risk [79]. Besides blood cell interaction effects, we also identified interaction effects between rs7487166 (chr12:132399243) and BMI that were associated with DNA methylation levels at cg13235087 (chr12:132383620). Cg13235087 is an EWAS signal for age and ethnicity, while rs7487166 is not a known GWAS signal, but both cg13235087 and rs7487166 are located in gene *ULK1*. *ULK1* regulates the mitophagy in adipocytes [80], and is associated with the risk of diabetic nephropathy [81] and obesity cardiomyopathy [82]. The G-BMI interaction association with DNA methylation at cg13235087 in *ULK1* suggests

that *ULK1* downstream metabolic profiles may be dependent on BMI levels, in addition to genetic background. These disease-related examples could provide more detailed insights into aspects of the underlying disease mechanisms that may be affected by genotype and exposure, or cellular and organismal environment interactions. Therefore, vmeQTLs findings could improve disease prediction, diagnosis, and prognosis.

A previous study by Ek et al. [38] explored evidence for local vmeQTL effects in whole blood in humans. Ek et al. [38] applied Bartlett's test to 729 independent samples and observed several hundred CpGs affected by vmeQTLs, of which several were identified to capture GxE interactions. Direct comparison of our results with those from Ek et al. [38] highlights 10 CpGs that were identified by both studies as vCpGs (Additional file 2: Table S8). Across the 10 CpGs, 3 CpGs (cg01701819, cg14859874, and cg15986326) are all previously identified EWAS signals of three immune diseases, Crohn's disease, inflammatory bowel disease, and ulcerative colitis. Two of the CpGs are EWAS signals for cancer, cg15986326 is associated with leukemia and cg01904145 is a EWAS signal of colorectal laterally spreading tumor. Four of the 10 vCpGs captured 7 G-cell type interactions in our data. However, the study by Ek et al. [38] also had some limitations, including high false positive rate for Bartlett's test [23, 39, 40] and relatively modest power for detecting vmeQTLs.

The majority of previous studies applying vQTL detection to human biological data have focused on gene expression and protein levels [13, 17, 83, 84], identifying common genetic variants that impact variability at these different biological layers. These studies have either analysed unrelated samples or took related samples as individuals where the family structure is regressed out. One hurdle in vQTL detection is that genetic variants in LD with mean-controlling QTLs may appear as spurious vQTL signals, especially if there are hidden mean-controlling QTLs [43]. The MZ twin design used in our study has advantages in this respect, because MZ twins are matched for genetic background, and many potential confounders such as age and sex. This aspect reduces noise and the identical germline genetic variation specifically avoid spurious vQTL signals arising from LD with hidden mean-controlling QTLs.

The current study also has several limitations. First, the MZ-twin based approach will miss interaction effects between genetic variants and age, sex, or other genetic variants (epistasis) because such effects would not lead to intra-MZ-twin-pair phenotype differences. Second, we only considered CpG sites for inclusion in this study if there was no missing methylation data across all twin pairs. As a result, we excluded approximately 40% of the Illumina 450K CpGs in the discovery set. This approach was taken because methylation data imputation would change the data distribution, which will affect DNA methylation variation. Third, blood cell type proportions in the current study are estimated using the Houseman algorithm [85] which is based on DNA methylation profiles. Although correlations between the estimated blood cell fractions using Houseman algorithm [85] and measured cell proportions are high, they are not perfect. A more precise estimation of blood cell subtypes [86] or direct measurement of blood cell subtype, will be helpful to capture GxE interactions of vmeQTLs in the future. Furthermore, there are additional confounders in our study, for example, aspects of the twins' external environment, as well as their chorionicity, for which we do not have reliable data. Previous studies have suggested

that chorionicity may affect DNA methylation levels in blood [103, 104], and it is also possible that the MZ twins have shared hematopoietic stem cells during development [105]. Moreover, it is very likely that environmental exposures are reflected to some extent in the human methylome, but the current study is limited by the scope of available data to robustly capture and study such external environmental exposures over the life course. Additionally, the BMI measurement date and blood sampling date were not concurrent in TwinsUK, but they were contemporaneous in NTR. Our previous simulation study found that power of vmeQTL detection may be lower without regressing out covariates first [40]. Therefore, all of these potential confounders may reduce vmeQTL detection power.

Several previous studies have considered how DNA methylation patterns in MZ twins compared to those from unrelated individuals. For example, a previous study proposed that the epigenetic profiles in MZ twins show greater similarity than expected at specific genomic regions, such as metastable epialleles (MEs), potentially due to the timing of the establishment of methylation marks prior to embryo cleavage [55]. On the other hand, a more recent study has also characterised many genomic regions where unrelated individuals and MZ twins display similar levels of methylation variability [87]. In our study we focused on genetic effects on intra-twin discordance in DNA methylation, making it unlikely that the detected vmeQTLs are MEs. We have compared our results with previous detected MEs and none of the vCpGs overlap with MEs. Furthermore, we validated 90% of vmeQTL-vCpG associations in non-twin individuals, which underscores their prevalence and independence of potential twin-specific features.

While vQTL detection for identification of GxE interactions is gaining popularity, detection of these effects on DNA methylation (vmeQTLs) is still in its infancy. Our study provides a first complete *cis* and *trans* analysis of these effects in nearly 2,000 MZ twins. However, further work is needed, for example, to dissect tissue-specific or tissue-shared patterns of vmeQTLs. We validated some of the blood-based vmeQTL-vCpG associations in adipose tissue, but were limited by the modest sample size of adipose tissue DNA methylation data. Furthermore, we identified potential GxE interactions with cellular environment, smoking, and BMI, but ideally future work would also explore other environmental factors that we have not considered in this work. Our initial analyses are conducted using the 450K array, but the EPIC arrays cover more functionally relevant CpG sites and are currently widely used in DNA methylation studies. Detecting vmeQTLs with larger sample sizes, across tissues, and using more recent and extensive methylation measurement technologies, would help gain a more comprehensive understanding of these effects.

## Conclusions

In summary, we identified and replicated genetic variants that affect the variability of DNA methylation in blood, using MZ twins. We validated these effects longitudinally, in adipose tissue, and in non-twin samples. The signals capture gene-environment interactions mostly with blood cell subtypes. Integrating our signals with results from GWAS and EWAS shows that vmeQTL-vCpG associations can provide novel insights into human complex traits.

## Methods

### vmeQTL simulation design in monozygotic twins

To gauge the most optimal test statistic to identify vmeQTLs in a MZ twin based study design, we initially carried out simulations to assess the performance of six vmeQTL detection models. In the simulations we assume that DNA methylation of each twin individual (Y) is influenced by a single genetic variant (G), environmental factors (E), gene-by-environment interaction (GxE), and noise (Eq. 1.1). The environmental factors include age (Uniform (20, 70)), sex (Binomial (1, 0.5)), BMI (Normal (25,1) for 80% of MZ twin pairs and BMI ~ Normal (25,3) for 20% of MZ twin pairs), and smoking status (Binomial (1,0.1), where 0 and 1 refer to two different smoking status levels of each twin). We assume that a G-smoking interaction influences DNA methylation variability. Noise is generated from the Normal, Chi-squared (df=1), and Gamma (shape1=2, shape2=0.5) distributions to model different methylation data distributions. We rescaled each term from 0 to 1 (G, E, GxE, and error in Eq. 1.1) with min–max scaling to keep the shape of their original distributions. The simulations were performed for different values of the coefficients of the additive genetic main effect ($a_G$) and for the gene-environment interaction effect ($a_{GxE}$), where the environmental main effect was set to $a_E > 0$, and with a constant $a_{error} = 0.2$. In each simulation setting we set the four coefficients to sum up to 1 ($a_G + a_{GxE} + a_E + a_{error} = 1$). Therefore Y, which represents the DNA methylation beta value at a single CpG site, ranges from 0 to 1. We also transformed the simulated beta value to a DNA methylation M value (M = log2(beta/1-beta)) to evaluate the models' performance when applying M values.

We assumed that there are altogether 350 MZ twin pairs and that the minor homozygous genotype group contains at least 5 twin pairs. The performance of the models was estimated under the scenarios where there is a GxE interaction ($a_G > = 0$, $a_{GxE} > 0$) or not ($a_G > = 0$, $a_{GxE} = 0$). The values of $a_G$ and $a_{GxE}$ were set to {0, 0.005, 0.01, 0.03, 0.05, 0.1, 0.15, 0.2, 0.25, 0.3, 0.35, and 0.4}.

To control for main effects of the environmental factor on DNA methylation, we adjusted the DNA methylation differences within MZ twins on their age, sex, and the difference in BMI and smoking status (0 and 1 refer to co-twins who are of the same or opposite smoking status, Eq. 1.2). Similarly, we adjusted the mean DNA methylation of MZ twins on age, sex, average BMI, and smoking status difference (Eq. 1.3). We assessed 6 models to identify vmeQTL, with or without including the mean level of DNA methylation as a covariate (Eqs. 1.4–1.9). We repeated the simulations 1000 times and calculated how often each model could detect a nominally significant ($P=0.05$) genotype effect on the intra-twin pair methylation difference. The frequencies correspond to the discovery rate when there is a GxE interaction, or to the false positive rate when there is no GxE interaction.

$$Y = \alpha_G \times G + \alpha_E \times E + \alpha_{GxE} \times G \times E + \alpha_{error} \times \text{error} \tag{1.1}$$

$$\text{res(abs)} : \text{residuals}\left(|\text{twin}_1 - \text{twin}_2| \sim \text{age} + \text{sex} + \text{diff}(\text{BMI}) + \text{diff}\left(\text{smoking}\right)\right) \tag{1.2}$$

$$\text{res(mean):residuals}(|\text{twin}_2 + \text{twin}_2|/2 \sim \text{age} + \text{sex} + \text{mean(BMI)} + \text{diff(smoking)}) \tag{1.3}$$

$$\text{Twins1}: \ \text{res(abs)} \sim \ \text{genotype} \tag{1.4}$$

$$\text{Twins2}: \ \text{res}^2(\text{abs}) \sim \ \text{genotype} \tag{1.5}$$

$$\text{Twins3}: \text{res(abs)/res(mean)} \sim \text{genotype} \tag{1.6}$$

$$\text{Twins4}: \ \text{res}^2(\text{abs})/\text{res}^2(\text{mean}) \sim \ \text{genotype} \tag{1.7}$$

$$\text{Twins5}: \ \text{res(abs)} \sim \ \text{genotype} + \text{res(mean)} \tag{1.8}$$

$$\text{Twins6}: \text{res}^2(\text{abs}) \sim \ \text{genotype} + \text{res(mean)} \tag{1.9}$$

### Monozygotic twin samples

We identified blood vmeQTLs in MZ twins from the TwinsUK cohort and replicated the results in MZ twins from the Netherlands Twin Register cohort (NTR). TwinsUK is the biggest adult twin registry in the UK [54]. DNA methylation data were generated for both MZ and dizygotic (DZ) twins. The discovery stage involved the analysis of 355 pairs of European-ancestry female monozygotic twins who were not selected for particular diseases, with the exception of 9 pairs of twins discordant for rheumatoid arthritis [101]. Twins included in this study had available genotype data, as well as complete information on age, sex, BMI, and smoking from TwinsUK [88]. The participants ranged from 18.6 to 79.9 (median = 58.8) years old, and their differences in body mass index (BMI) ranged from 0 to 19.15 (median = 1.89). Smoking status was categorized as current smokers and non-current smokers (including ex- and non-smokers), and 316 pairs of MZ twins were of the same smoking status at the sample collection time, while 39 pairs were discordant for smoking status. NTR is a population-based cohort of over 200,000 people from across the Netherlands [89]. It consists of twin-families, i.e. twins, their parents, spouses, and siblings aged between 0 and 99 years at recruitment and started around 1987 with new-born twins, and adolescent and adult twins. DNA methylation profiles were derived for both MZ and DZ twins. We only included the MZ twins who have available genotype and DNA methylation data, age, sex, smoking, and cell component information. In families with multiple MZ twin pairs, we randomly selected one MZ twin pair to include. Individuals identified as non-European based on genotype principal component analysis (PCA) were excluded. As a result, we replicated the identified vmeQTL-vCpG

associations in 633 pairs of MZ twins from the NTR, comprising 442 pairs of female twins and 191 pairs of male MZ twins. The age of the NTR participants ranged from 18 to 78.6 years old with a median 32.5 years old. The differences of BMI within the NTR MZ twin pairs were from 0 to 12.1 with a median value of 1.31. Altogether, 534 pairs of NTR MZ twins were at the same smoking status while 99 pairs of MZ twins were not. Informed consent was obtained from all participants.

### Genotype data

In the TwinsUK cohort, genotype data were generated by a combination of Illumina HumanHap300, HumanHap610Q, 1 M-Duo and 1.2MDuo 1 M arrays. Genotype imputation was based on the 1000 genome phase 3 version 5 with the Michigan imputation server. More details of genotype imputation and processing of all data from TwinsUK have been previously described [54]. In the sample of 355 twin pairs from TwinsUK, we excluded SNPs that showed Hardy–Weinberg equilibrium $P$-values less than $10^{-6}$, had minor allele frequency (MAF) less than 0.05, or imputation quality info lower than 0.3. We also removed SNPs where there were fewer than 5 twin pairs in any of the genotype groups, to avoid other potential biological scenarios such as genomic imprinting. In the NTR cohort, genotyping was done on the Illumina GSA-NTR, Affymetrix Axiom-NTR and 6.0 arrays following manufacturers protocols and white papers. The 3 platforms were then separately imputed with Beagle 5.4 to the 1000 genomes phase 3 version 5 panel. To replicate the vmeQTL-vCpG associations, we also removed SNPs with 1/lower than $10^{-6}$ Hardy–Weinberg equilibrium $P$-values, 2/a MAF less than 0.05, 3/fewer than 5 pairs of twins in any genotype group. We used best guess genotypes and human reference genome build 37 (HG19) to assemble genomic positions in both TwinsUK and NTR genotype data.

### DNA methylation profiling and processing

DNA methylation data in TwinsUK blood samples were measured by the Infinium HumanMethylation450 BeadChip array (Illumina 450K), as previously described [88]. The DNA methylation data were processed by the Exponential-Normal mixture distribution (Enmix) R package to obtain DNA methylation beta values [90]. Specifically, background correction was carried out with out-of-band type I probe intensity to model the background noise, dye-bias correction was conducted with Regression on the Logarithm of Internal Control probes (RELIC) method, and adjustment of probe design was performed with the Regression on Correlated Probe (RCP) method. We removed multimapping, sex-chromosome, and blacklisted probes. We also excluded probes with any missing values to avoid introducing bias into the data distribution from missing data imputation, resulting in the detection of spurious vmeQTLs. Altogether, 261,793 probes were ultimately tested in the TwinsUK discovery set.

DNA methylation data in NTR blood samples were assessed with the Infinium HumanMethylation450 BeadChip Kit by the Human Genotyping facility (HugeF) of ErasmusMC, the Netherlands (http://www.glimdna.org/) as part of the Biobank-based Integrative Omics Study (BIOS) consortium [60]. DNA methylation measurements have been described previously [60, 91]. Sample-level QC was performed using MethylAid [92]. Probes were set to missing in a sample if they had an intensity value of exactly

zero, or a detection P > 0.01, or a bead count is less than 3. After these steps, probes that failed based on the above criteria in more than 5% of the samples were excluded from all samples (only probes with a success rate higher than 0.95 were retained). The following probes were also removed: probes on sex chromosomes, probes with a SNP within the CpG site (at the C or G position) irrespective of MAF in the Genome of the Netherlands (GoNL) population, and ambiguously mapping probes reported by Chen et al. with an overlap of at least 47 bases per probe [93]. The methylation data were normalized with functional normalization [94].

### DNA methylation data adjustment

We carried out the same DNA methylation data covariate adjustment procedure in TwinsUK and NTR, except that sex was an additional covariate in NTR. The DNA methylation data were first regressed on age, BMI, smoking, technical batch, and blood cell composition, where each MZ twin pair was the unit of analysis (Eqs. 2.1–2.2). Smoking status was binary to represent whether the smoking status of each twin pair is concordant or not, where ex- and non-smokers are taken as the same status. The technical batch represents whether co-twin samples were run on the same array or not. Because measured cell counts are not available in TwinsUK, cell proportions were estimated by the Houseman et al. [85] algorithm in both TwinsUK and NTR cohort samples. Cell estimates were obtained for plasmablasts, memory and effector T cells (CD8pCD-28nCD45Ran), naïve CD8 + T cells, naïve CD4 + T cells, CD8 + T cells, CD4 + T cells, natural killer cells, B cells, monocytes, and granulocytes. To assess the performance of the blood cell proportion estimation, we compared the correlation between the estimated cell proportions and previously measured cell proportions from the same samples in NTR [102]. Briefly, in the NTR cohort, a haematological profile was obtained from EDTA blood using the Coulter system (Coulter Corporation, Miami, USA) after the samples arrived in the laboratory. Parameters provided included white blood cell count (WBC), with percentages and numbers of neutrophils, lymphocytes, monocytes, eosinophils, basophils. Correlations between parameters that reflect the same estimated and measured white blood cell fractions are high (Additional file 2: Table S9). For example, the Pearson's correlation between estimated granulocytes and measured neutrophils is 0.90, and the estimated and measured monocytes correlation is 0.71 (Additional file 2: Table S9). To maintain consistency in covariates across cohorts, we used estimated cell proportions in both TwinsUK and NTR. To reduce collinearity across different cell subtypes, we calculated correlations across different cells in the TwinsUK dataset, and only included cell types that showed the lowest correlation between each other. As result, we included naïve CD8 + T cells, CD8 + T cells, monocytes, and granulocytes as covariates in both cohorts (Additional file 1: Fig. S4).

After regressing out effects of covariates on the intra-twin difference on DNA methylation (Eq. 2.1) and average DNA methylation level (Eq. 2.2), the residuals and genotypes were fit into a linear model to detect vmeQTLs (Eq. 2.3) and meQTLs (Eq. 2.4), respectively.

$$\text{DNAm}(|\text{twin}_1 - \text{twin}_2|) \sim \text{age} + \text{diff}(\text{BMI}) + \text{diff}(\text{smoking}) + \text{diff}(\text{cell}) + \text{diff}(\text{batch}) \tag{2.1}$$

$$\text{DNAm}(\frac{|\text{twin}_1 + \text{twin}_2|}{2}) \sim \text{age}+\text{mean(BMI)}+\text{diff}\big(\text{smoking}\big)+\text{mean(cell)}+\text{diff(batch)}$$

(2.2)

$$\text{vmeQTL}: \text{ Residuals}(2.1) \sim \text{ Genotype}$$

(2.3)

$$\text{meQTL}: \text{ Residuals}(2.2) \sim \text{ Genotype}$$

(2.4)

### vmeQTL and meQTL detection

For each tested CpG site, we estimated both the genetic effects on DNA methylation variance (vmeQTLs) and mean level (meQTLs). We considered both local and distal genetic effects, where local effects (in *cis*) are cases where the SNP is within 1Mbp of each side (*cis*) around each CpG. SNPs outside of *cis* regions are defined as distal (*trans*) QTLs for the CpG. We performed *cis*-QTL identification with TensorQTL [45], which applies the permutation-based threshold for each CpG via the beta-approximation method [45]. Briefly, we conducted 10,000 permutations for each CpG. We estimated associations between each CpG and SNP in the *cis* window and saved the most significant association (P(true)) for each CpG. We then permuted the data and estimated associations between the permuted DNA methylation data and SNPs in the *cis* window, and kept the most significant *P* value in each permutation (10,000 *P*-values in total for each CpG). We then fitted 10,000 *P*-values of each CpG into a beta distribution, and assigned the cumulative density from 0 to P(true) in this beta distribution as P(beta). Finally, we corrected for multiple testing using P(beta) from all CpGs to observe Pt(beta) (Storey qvalue < 0.05). CpGs with a P(beta) < Pt(beta) have significant *cis*-QTLs and P(beta) is the multiple testing significance threshold for that specific CpG. We used this procedure to obtain *cis*-meQTLs and *cis*-vmeQTLs for each CpG, as well as in the conditionally independent analysis with TensorQTL.

We used matrixEQTL to identify *trans*-meQTLs and *trans*-vmeQTLs [46]. Here we performed 20 permutations for each SNP-CpG pair and used these to determine the multiple testing threshold via FDR = (N(P(permutation) < threshold)/N(P(real) < threshold))/20. The thresholds obtained were $P < 1.08e{-}11$ for *trans*-vmeQTLs and $P < 4.5e{-}8$ for *trans*-meQTLs when assigning FDR less than 0.05. For each CpG, we conducted LD clumping of all the trans QTL results using Plink [95] with a window of 500 kb and a threshold $r^2 = 0.2$. Only one index trans-vmeQTL remained for each CpG.

### QTL sensitivity analyses

We conducted additional analyses to filter out potential spurious vmeQTL signals. First, we excluded previously detected imprinted CpGs from the analysis, because we wanted to focus on identification of CpGs affected by GxE interactions. Second, we carried out analyses to try to filter out spurious signals that may arise from the correlation between the mean and the variance in DNA methylation at a CpG site. For each CpG, we carried out LD clumping between the meQTL and vmeQTL using Plink [95], and this was based on the vmeQTLs and meQTLs *P*-values, with a 500 kb window and a threshold of $r^2 = 0.2$ [95]. LD clumping was carried out separately: 1) for *cis*-vmeQTLs and *cis*-meQTLs, 2) for *trans*-vmeQTLs and *trans*-meQTLs, 3) for *cis*-vmeQTLs and

*trans*-meQTLs, and lastly 4) for *trans*-meQTLs and *cis*-meQTLs. We only include the most significant QTL in each LD block and clumped QTLs were removed. Third, for those SNPs that were identified as both a meQTL and vmeQTL for the same CpG, we carried out additional regression analyses. We regressed out the meQTL effect on DNA methylation of each twin individual (Eq. 3.1) and evaluated whether the genetic variant still affects the intra-twin discrepancies in DNA methylation (Eqs. 2.1, 2.3). Finally, we conducted a sensitivity analysis with two additional covariates, average BMI and cell proportion within twin pair, to confirm that the covariate adjustment did not influence vmeQTL identification (Eq. 3.2).

$$\text{DNAm} \sim (1|\text{fam}) + \text{meQTL} \tag{3.1}$$

$$|\text{twin}_1 - \text{twin}_2| \sim \text{age} + \text{diff(BMI)} + \text{diff}\big(\text{smoking}\big) + \text{diff(cell components)}$$
$$+ \text{diff(batch)} + \text{mean(BMI)} + \text{mean(cell components)} \tag{3.2}$$

### Genome annotations and functional enrichment analysis

We explored the genomic distribution of vmeQTLs, mQTLs, and the corresponding CpG sites that they associate with. We downloaded genic regions from the NCBI Reference Sequence collection (RefSeq), and CpG island density and transcription factor binding site (TFBS) data for the GM12878 cell line from ENCODE 3 in the UCSC table browser (https://genome.ucsc.edu/cgi-bin/hgTables). We obtained chromatin accessibility data from Epigenetic Roadmap (https://egg2.wustl.edu/roadmap/data/byFileType/chromhmmSegmentations/ChmmModels/imputed12marks/jointModel/final/). Enrichment analysis was carried out using an adapted version LOLA R package [96], which uses a two-tailed Fisher's exact test [96]. We estimated the enrichment of QTLs and CpGs in specific genomic regions, where each annotation region had to contain at least 10 QTL or CpG signals in order to be considered in the analysis.

To explore pathways or molecular functions related to the genes that the vCpGs annotated to, we carried out two additional analyses. First, we carried out functional enrichment analysis of vCpGs in GoMeth [97] using the ∼ 261,000 tested CpGs as background CpGs. Second, we performed gene set enrichment analysis of the genes that the vCpGs annotated to, using the genes near to the ∼ 261,000 tested CpGs as the background set. The gene set enrichment analysis was performed by FUMA [59], and we focused on enrichments in the Gene Ontology (GO) and Kyoto Encyclopedia of Genes and Genomes (KEGG) datasets.

### Gene-environment interaction analysis

To explore whether vmeQTL-vCpG associations capture GxE interaction, we carried out direct tests of GxE interactions at the level of the individual twin. The GxE interaction analysis was performed using 7 environmental or modifier factors (E), which included BMI, smoking, naïve CD8 T cells, CD8 T cells, granulocytes, monocytes, and myeloid and lymphoid ratio (MLR). MLR was calculated as the ratio of monocytes over the sum of B cells, CD8 T cells, CD4 T cells, and NK, and has previously been linked to diseases such as haematological malignancy [98] and infectious diseases such as malaria [99].

In the GxE interaction analysis, we took a two-stage approach. We first regressed out DNA methylation levels on relatedness, batch effects, age and five environmental factors excluding both the factor of interest and MLR, in a linear mixed model (Eq. 4.1). The residuals from this model were then tested for GxE interactions with the specific environment factor of interest, and we assessed the significance of interaction effects (Eq. 4.2). This approach was used to test for interactions with six of the seven environmental factors, specifically BMI, smoking, naïve CD8 + T cells, CD8 + T cells, granulocytes, and monocytes. When testing for G-MLR interactions, in the first stage we only adjusted for the sample relatedness, batch effects, age, BMI, smoking, and 4 cell components, and then took the residuals to the second stage interaction analysis. We only report GxE interactions that surpassed FDR < 0.05 after multiple testing corrections.

$$\text{DNAm} \sim (1|\text{fam}) + (1|\text{batch}) + \text{age} + \text{other covariates} \tag{4.1}$$

$$\text{residual(DNAm)} \sim \text{vmeQTL} + \text{E} + \text{vmeQTL} \times \text{E} \tag{4.2}$$

for example, when testing vmeQTL-BMI interaction, Eq. 4.1 would be $\text{DNAm} \sim (1|\text{fam}) + (1|\text{batch}) + \text{age} + \text{smoking} + \text{CD8T.naive} + \text{CD8T} + monocytes + granulocytes + \text{MLR}$, and Eq. 4.2 would be $\text{residual(DNAm)} \sim \text{vmeQTL} + \text{BMI} + \text{vmeQTL} \times \text{BMI}$

### vmeQTL-vCpG comparison with Blueprint data cell-type meQTLs

To assess whether the vmeQTL-vCpG effects that capture G-cell type interactions in our data may be cell-type meQTLs, we compared out results with blood cell-type meQTLs obtained from the Blueprint data. We obtained the CpGs associated with meQTLs in monocytes and in T cells from the Blueprint dataset (http://blueprint-dev.bioinfo.cnio.es/WP10/qtls) [44]. The Blueprint project recruited blood donors from the UK population. Monocytes were obtained from 200 donors and CD4+T cells were obtained from 169 donors. The DNA methylation data were generated by Infinium HumanMethylation450 assays. When comparing results, we compared whether the vCpGs for which the vmeQTL showed evidence for SNP-monocyte interactions, were also mCpGs in monocytes in Blueprint data. We used the same approach for T cells, except for comparisons to the T cell mCpGs in the Blueprint dataset we used the set of vCpGs that were influenced by vmeQTL-CD8+T cells and vmeQTL-naïve CD8+T cells in our data.

### Longitudinal stability of vmeQTL effects

To assess whether vmeQTL exhibit longitudinal stability, we explored a data subset of MZ twins from the TwinsUK sample that had longitudinal blood DNA methylation profiles. Altogether 56 pairs of MZ twins from the TwinsUK discovery set had a follow-up sample ranging from 4 to 16 years from baseline. The DNA methylation profiles of the first-time point blood samples were generated by the Infinium HumanMethylation450 BeadChip (450K) and also using the Infinium MethylationEPIC array (EPIC), but the second-time point blood samples only had EPIC data. Therefore, we used EPIC data for longitudinal stability exploration.

After regressing out effects of sample relatedness, batch effect, age, BMI, smoking, and 4 cell types, we compared the residuals of DNA methylation data profiled by 450K and

EPIC to estimate the correlation between the two arrays. We excluded vCpGs with a correlation lower than 0.2 (Spearman's rank correlation). Due to the lower sample size, we also excluded some genetic variants, specifically we excluded vmeQTLs with MAF < 5% in the subset of 56 MZ twin pairs, and also SNPs where there were fewer than 5 MZ twin pairs in any of the genotype groups. This resulted in a total set of 223 vmeQTL-vCpG associations that were assessed longitudinally in the EPIC MZ twin dataset.

To estimate longitudinal stability, we first estimated whether the intra-twin difference in DNA methylation increases with longitudinal sample collection time interval. We regressed out the covariates (Eq. 2.1) on the intra-twin DNA methylation difference, and estimated the association between methylation discrepancies in the intra-twin difference at the two time points and time interval. Second, we also assessed evidence for longitudinal stability by directly estimating vmeQTL effects on the DNA methylation intra-twin differences at both time points using EPIC data in the subset of 56 MZ twin pairs (Eq. 2.3). We considered effects to validate where the vmeQTL signal was significant after correction for multiple testing (FDR < 0.05).

### Validating blood vmeQTLs in adipose tissue

To explore whether vmeQTLs can be tissue-shared, we explored their effects in adipose tissue. Adipose tissue samples were obtained from 85 pairs of MZ twins from the TwinsUK cohort [100], of which 56 MZ twin pairs also had blood samples in the TwinsUK discovery set in this study. Adipose samples were collected from female twins (age range 41.5 to 84.6 years, median age 59; intra-twin difference in BMI range from 0 to 10, median BMI difference 1.75). Smoking status categorisation was applied using the same approach as in the discovery blood analysis, and only 3 MZ twin pairs had a discordant smoking status in the adipose dataset. To validate the blood vmeQTL-vCpG associations in adipose tissue, we used the same method as that in blood vmeQTL identification. Specifically, age, batch effect, smoking, BMI, microvascular endothelial cells, adipocytes, and macrophages were included as covariates. In considering tissue-specific and shared effects we excluded vmeQTLs where the SNP had < 5% MAF in the adipose tissue dataset. We also excluded SNPs that had fewer than 5 MZ twin pairs in any genotype category. Altogether, we assessed associations at 207 vmeQTL-vCpG pairs in adipose tissue. We set FDR < 0.05 as the threshold to validate the candidate vmeQTL-vCpG associations.

### Replication of vmeQTLs in non-twins

To assess whether the twin-based vmeQTLs could also be detected in non-twin samples, we carried out analyses using DNA methylation profiles from individuals in the MRC National Survey of Health and Development (NSHD), or the UK 1946 birth cohort. The NSHD analyses were performed in 1,348 individuals at age 53, comprising of 704 female and 644 males, and 330 current smokers. DNA methylation data in the NSHD cohort were profiled by the EPIC array, as previously described [49], and genotype imputation and DNA methylation data processing followed the same procedures [49]. In this study, we excluded SNPs where there were fewer than 5 samples in any of the genotype groups. Altogether, we assessed 200 associations in the NSHD cohort. To perform vmeQTL analysis in this sample of unrelated individuals, we initially regressed out covariates

including sex, smoking, and cell proportions from the DNA methylation levels at the candidate vCpG. We then applied DRM [39] to estimate the vmeQTL effects. We also tested genetic impacts using SVLM [58] as a sensitivity analysis in the NSHD dataset. Briefly, DRM estimates the distance of samples to the median value of each genotype group and then fits it and the genotype into a linear model. SVLM regresses out the genetic variant effects and then takes the squared residuals and genetic variants into a linear model. We considered results to replicate if the associations surpassed FDR < 0.05 threshold for both DRM and SVLM.

### Comparison of vmeQTLs and vCpGs with EWAS and GWAS signals

We compared the vmeQTLs to previously published GWAS signals available from the GWAS catalog (https://www.ebi.ac.uk/gwas/, downloaded in August 2023).

We also compared vCpGs to previously published EWAS signals. Here, we combined previously published results available from the EWAS catalog (http://ewascatalog.org/, downloaded in August 2023) and EWAS atlas (https://ngdc.cncb.ac.cn/ewas/atlas, downloaded in August 2023). We then estimated the enrichment of vmeQTLs to fall in GWAS loci of a particular phenotype or disease, and we also clustered phenotypes into categories where appropriate. We used the same approach for vCpGs and EWAS. To estimate the enrichment of vmeQTLs and vCpGs in each phenotype (category), we excluded the GWAS and EWAS terms with less than 10 vmeQTL and vCpG signals. To assess enrichments or depletion in each set of genomic loci, we performed a two-tailed Fisher's exact test.

### Supplementary Information

Additional file 1. Supplementary methods and results, supplementary figures S1-S6, and cohort-specific acknowledgements.

Additional file 2. Supplementary tables S1-S10.

Additional file 3. Review history.

#### Acknowledgements
The authors express their gratitude to the research volunteers who participated in the study. Cohort-specific acknowledgments are provided in Additional file 1.

#### Peer review information

#### Review history
The review history is available as Additional file 3.

#### Authors' contributions
XZ and JTB designed the project. XZ analysed the project, and JTB supervised the project. XZ, JTB, IY, SV, JC-F, MM, and P-CT provided data and analysis for the TwinsUK datasets. HM, AW, KKO provided data and analysis for the NSHD cohort. DB, JvD, JJH provided data and analysis for the NTR cohort. JM and MF helped with results interpretation. All authors wrote and approved the final version of the manuscript.

#### Funding
This work is funded in part by the UK ESRC (ES/N000404/1 to JTB). XZ acknowledges the financial support of the China Scholarship Council (CSC).

#### Data availability
The code for carrying out the analysis is available at GitHub (https://github.com/ZXiaopu/vmeQTLs) [106] and has been deposited in Zenodo under the DOI: https://doi.org/10.5281/zenodo.18031615 under an MIT license [107].
The TwinsUK HumanMethylation450 BeadChip data are available via GEO under accession numbers GEO GSE62992 and GSE121633 (blood methylation). Access to the TwinsUK genotype data can be applied for through the cohort data

access committee, see https://twinsuk.ac.uk/resources-for-researchers/access-our-data/. The HumanMethylation450 BeadChip data from the NTR are available as part of the Biobank-based Integrative Omics Studies (BIOS) Consortium in the European Genome-phenome Archive (EGA), under the accession code EGAD00010000887, https://ega-archive.org/datasets/EGAD00010000887 (Study ID EGAS00001001077, Title: The mission of the BIOS Consortium is to create a large-scale data infrastructure and to bring together BBMRI researchers focusing on integrative omics studies in Dutch Biobanks, contact: The BIOS Consortium: Biobank-based Integrative Omics Studies, Contact person: Rick Jansen). All NTR data can be requested by bona fida researchers (https://ntr-data-request.psy.vu.nl/). The pipeline for DNA methylation-array quality control developed by the Biobank-based Integrative Omics Study (BIOS) consortium is available here: https://molepi.github.io/DNAmArray_workflow/ copy archived at Sinke, 2020 (https://doi.org/10.5281/zenodo.3355292). The 1946BC methylation dataset is available in the public domain through https://doi.org/10.5522/NSHD/S202. Access to further individual-level data can be applied for through the cohort data access committee, via https://nshd.mrc.ac.uk/data-sharing/.

## Declarations

### Ethics approval and consent to participate

All research participants have signed informed consent prior to taking part in any research activities. Ethical approval for TwinsUK was granted by the National Research Ethics Service London-Westminster, the St Thomas' Research Ethics Committee (REC reference numbers: EC04/015 and 07/H0802/84). The NTR study was approved by the Central Ethics Committee on Research Involving Human Subjects of the VU University Medical Centre, Amsterdam, an Institutional Review Board certified by the U.S. Office of Human Research Protections (IRB number IRB00002991 under Federal-wide Assurance- FWA00017598; IRB/institute codes, NTR 03-180). Ethical approval for 1946BC was granted by the Central Manchester Research Ethics Committee (07/H1008/168 and 07/H1008/245) and the Scotland A Research Ethics Committee (08/MRE00/12).

### Consent for publication

Written informed consent was obtained from all participants for the publication of their data.

### Competing interests

The authors declare no competing interests.

### Author details

[1]Department of Twin Research and Genetic Epidemiology, King's College London, London, UK. [2]Department of Bioinformatics, Graduate School of Health Sciences, Hacettepe University, Ankara, Turkey. [3]Department of Biological Psychology, Vrije Universiteit Amsterdam, Amsterdam, The Netherlands. [4]Amsterdam Reproduction and Development Institute, Amsterdam, The Netherlands. [5]Amsterdam Public Health Research Institute, Amsterdam, The Netherlands. [6]MRC Integrative Epidemiology Unit, University of Bristol, Bristol, UK. [7]MRC Unit for Lifelong Health and Ageing, Institute of Cardiovascular Science, University College London, London, UK. [8]MRC Epidemiology Unit, University of Cambridge School of Clinical Medicine, L3 Institute of Metabolic Science, University of Cambridge, Cambridge, UK. [9]Department of Paediatrics, University of Cambridge, Cambridge, UK.

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

## 